# A Comparative Study of the Antiviral Properties of Thermally Sprayed Coatings against Human Coronavirus HCoV-229E

Elnaz Alebrahim [1,†], Hediyeh Khatibnezhad [1,†], Morvarid Mohammadian Bajgiran [1], Magan Solomon [2], Chen Liang [2,3], Selena M. Sagan [3,4], Rogerio S. Lima [5], Jörg Oberste Berghaus [5], Maniya Aghasibeig [5] and Christian Moreau [1,*]

1 Department of Mechanical, Industrial and Aerospace Engineering, Concordia University, 1455 de Maisonneuve Blvd. W, Montreal, QC H3G 1M8, Canada
2 Department of Medicine, Division of Experimental Medicine, McGill University, 3755 Côte Ste-Catherine Road, Montreal, QC H3T 1E2, Canada
3 Department of Microbiology and Immunology, McGill University, 3775 Rue University Street, Montreal, QC H3A 2B4, Canada
4 Department of Biochemistry, McGill University, 3655 Promenade Sir William Osler, Montreal, QC H3G 1Y6, Canada
5 National Research Council Canada, 75 de Mortagne Blvd., Boucherville, QC J4B 6Y4, Canada
* Correspondence: christian.moreau@concordia.ca
† These authors contributed equally to this work.

**Abstract:** For decades, novel viral strains of respiratory tract infections have caused human pandemics and initiated widespread illnesses. The recent coronavirus disease 2019 (COVID-19) outbreak caused by the SARS-CoV-2 virus has raised an urgent need to develop novel antiviral coatings as one of the potential solutions to mitigate the transmission of viral pathogens. Titanium dioxide is considered an excellent candidate for viral disinfection under light irradiation, with the potential to be activated under visible light for indoor applications. This research assessed the antiviral performance of thermally sprayed $TiO_2$ coatings under UVA and ambient light. We also report the antiviral performance of $TiO_2$ composites with other oxides, such as $Cu_2O$ and $Al_2O_3$, produced by suspension plasma spray, atmospheric plasma spray, and suspension high-velocity oxygen fuel techniques. To evaluate the antiviral performance of the above coatings in a containment level-2 laboratory, a human common cold coronavirus, HCoV-229E, was initially used as a relevant surrogate for SARS-CoV-2. Coatings were also analyzed using SEM and XRD and were classified based on their surface roughness, porosity, and phase composition. Collectively, the thermally sprayed coatings showed comparable or slightly better antiviral activity compared to copper. The most significant level of activity observed was approximately 20% to 50% higher than that of a pure copper plate.

**Keywords:** $TiO_2$; $Cu_2O$; photoactivity; contact killing; antiviral coatings; thermal spray; human coronaviruses HCoV-229E; COVID-19

## 1. Introduction

As of March 2020, the widespread and rapid increase in the novel human coronavirus disease (COVID-19), which causes the severe acute respiratory syndrome (SARS) named SARS-CoV-2, has raised worldwide attention toward public health. The World Health Organization reported in June 2023 over 767 million cases of infection, including over 6.9 million deaths, due to the COVID-19 pandemic [1,2]. After SARS coronavirus and Middle East Respiratory Syndrome (MERS) coronavirus, this is the third highly pathogenic human coronavirus that was spread over the past two decades [1].

Vaccination against SARS-CoV-2 has been developed since 2020 with acceptable success [3]. However, vaccination cannot prevent the emergence of new variants. Infectious pathogens could be transmitted either directly from close person-to-person contact or

indirectly through hand-touch contact after deposition on surfaces [4]. Based on the literature, airborne transmission of SARS-CoV-2 virus was the main transmission route [5], but previous studies revealed that the highly contagious COVID-19 virus could remain infectious for days on some surfaces [4]. To inhibit the risk of indirect transmission of infections, coated surfaces with intrinsic antiviral materials could be more effective than frequent cleaning of the environment. Regular cleaning in public places is costly, and re-contamination may occur. An incomplete cleaning process or ineffective cleaning agents may also leave viral residues that may continue to contaminate surfaces [6,7]. Moreover, respiratory disease symptoms such as coughing and mucus production would contaminate the cleaned surfaces frequently and increase the risk of virus transmissions [1]. Accessories in healthcare facilities are mostly made of stainless-steel surfaces as a common material due to its corrosion resistance and clean appearance. However, stainless steel has no antimicrobial capabilities, and pathogens can attach and grow on it easily [8,9]. Thus, the deposition of highly efficient, low-cost, and environmentally friendly antiviral coatings on such surfaces is a reasonable solution to mitigate the transmission of viral diseases. The antimicrobial properties of various metals, particularly copper and silver, have been extensively studied and documented [10]. The COVID-19 pandemic caused a surge in research interest in copper due to its ability to deactivate viruses, which is shown to be directly related to the amount of copper ions released on the surface of the alloy.

Since the antibacterial properties of titanium dioxide ($TiO_2$) photocatalyst were reported by Matsunaga et al. in 1985 [11], $TiO_2$ has attracted ever-growing worldwide attention for antipathogenic applications due to its unique mechanical and chemical resistance [12–15]. $TiO_2$ can be excited by light irradiation to produce powerful reactive oxygen species (ROS) with strong oxidizing power for airborne pathogen inactivation under ambient conditions. Interaction of the virus with the photocatalytic surface results in substantial changes in the virus structure, leading to the loss of the ability of the virus to attack host cells [13,16].

Among the two main crystalline phases of $TiO_2$, the anatase phase presents the highest activity for photoactive applications. However, previous reports have shown the synergistic effects of the mixture of anatase and rutile phases that could enhance the photoactivity of $TiO_2$ photocatalyst [8,17–19]. Furthermore, based on the literature, increasing the surface area could also improve the photoactivity of $TiO_2$ photocatalysts by increasing the active sites for reaction initiation and decreasing the charge carrier's recombination rate by introducing some trapping sites [20–22].

Alumina ($Al_2O_3$) could be utilized with $TiO_2$ to improve the mechanical properties of $TiO_2$ coatings [23,24]. Among different alumina phases, the stable corundum phase ($\alpha$-$Al_2O_3$) presents high chemical stability, high electrical resistance, and good thermal and mechanical properties [25].

Copper oxide ($Cu_2O$) coatings also show potent antiviral activity, even under dark conditions [4,26,27]. Recent interest in visible light-sensitive $Cu_2O$-$TiO_2$ nanocomposite has been substantial due to its sustainable antiviral activity under dark conditions. $Cu_2O$-$TiO_2$ nanocomposite is desirable for indoor environmental remediation applications and also exhibits strong antiviral activity under dark conditions when indoor light instruments are switched off [28,29]. These coatings would be less susceptible to viral contamination and can minimize the spread of the viruses.

$TiO_2$ films and coatings have been fabricated by various techniques; however, introduction into the market of antiviral coatings relies on the capability of the coating deposition technique for mass production. Thermal spray technology appears as a versatile and rapid processing approach where the large surface area coverage and high deposition rate (up to hundreds of microns in a couple of minutes) make this deposition process compatible with the industry [8,30,31]. Another advantage of thermal spray processes is the flexibility of producing a wide variety of high-quality coatings on substrates with different shapes and sizes [32]. In addition, historically, copper has been more expensive than $TiO_2$. Furthermore, the price can vary depending on various factors, such as global demand, supply, economic

conditions, and market dynamics. Thus, using copper-based materials on a large scale during a pandemic could be more costly.

Based on our previous work [17,33], thermally sprayed sub-stoichiometric $TiO_{2-x}$ coatings presented remarkable photocatalytic performance under visible light. These coatings were deposited using the suspension plasma spray (SPS) process. The results indicated the effect of the oxygen deficiency and $Ti^{3+}$ ions on decreasing the bandgap energy and shifting the absorption edge to the visible light range [33].

In this research work, $TiO_2$, $Cu_2O$, $TiO_2$-$Cu_2O$, and $TiO_2$-$Al_2O_3$ antiviral coatings were produced using different thermal spray processes, including atmospheric plasma spray (APS), suspension plasma spray (SPS), and suspension high-velocity oxygen fuel (S-HVOF). To investigate the antiviral activity of the produced coatings, the human common cold coronavirus, HCoV-229E, was used as a surrogate for SARS-CoV-2 in a containment level-2 laboratory.

## 2. Results and Discussion

### 2.1. Phase Composition

XRD patterns and the derived information of $TiO_2$, $Cu_2O$, $TiO_2$-$Cu_2O$, and $TiO_2$-$Al_2O_3$ coatings are shown in Figure 1 and Table 1. According to the results, various ranges of anatase phase contents were produced in $TiO_2$ coatings, where T1-SPS and T5-SHVOF coatings presented the highest anatase content.

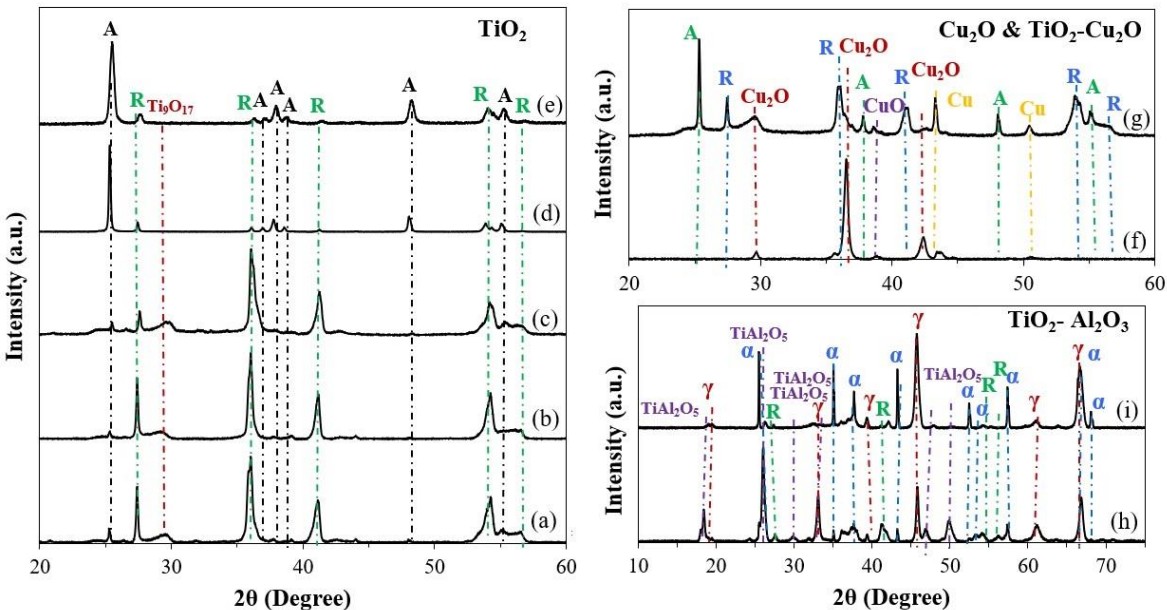

**Figure 1.** XRD patterns of coatings as (a) T2-APS, (b) T3-APS, (c) T4-APS, (d) T1-SPS, (e) T5-SHVOF, and (f) C-SPS, (g) TC-SPS, (h) A40%T-APS, (i) A13%T-APS, in which A denotes anatase phase, R denotes rutile phase, $\alpha$ is $\alpha$-$Al_2O_3$, and $\gamma$ is $\gamma$-$Al_2O_3$.

XRD patterns (Figure 1) show $Cu_2O$, $CuO$, and $Cu$ phases for C-SPS coating and anatase, rutile, $Cu_2O$, and $Cu$ phases for TC-SPS coating. According to the literature, copper (Cu) and $Cu_2O$ are considered antibacterial/antiviral materials by a direct contact mechanism [4,34,35]. On the other hand, the inferior antiviral activity of the CuO phase might be related to the less potential of the CuO phase to adsorb and denature proteins [4]. In $TiO_2$-$Al_2O_3$ coatings, some part of the active $TiO_2$ phase was transformed into the $TiAl_2O_5$ phase, as shown in Figure 1 and Table 1. The formation of the aluminum titanate phase could be detrimental to the photoactivity of the coatings.

**Table 1.** Phase content of the coatings derived from Figure 1.

| Sample | %Anatase | %Rutile | %Ti$_9$O$_{17}$ | %Cu$_2$O | %CuO | %Cu | %$\alpha$-Al$_2$O$_3$ | %$\gamma$-Al$_2$O$_3$ | %TiAl$_2$O$_5$ |
|---|---|---|---|---|---|---|---|---|---|
| T1-SPS | 83 | 17 | - | - | - | - | - | - | - |
| T2-APS | 9.5 | 64.5 | 26 | - | - | - | - | - | - |
| T3-APS | 6 | 67.5 | 26.5 | - | - | - | - | - | - |
| T4-APS | 28 | 72 | - | - | - | - | - | - | - |
| T5-SHVOF | 82 | 18 | - | - | - | - | - | - | - |
| C-SPS | - | - | - | 65.1 | 18.2 | 16.7 | - | - | - |
| TC-SPS | 42 | 19 | - | 34 | - | 5 | - | - | - |
| A40%T-APS | - | 6 | - | - | - | - | 32.5 | 37 | 24.5 |
| A13%T-APS | - | 2 | - | - | - | - | 55 | 32 | 11 |

### 2.2. Microstructure and Morphology

Three-dimensional (3D) maps of the top surfaces of the coatings taken by a confocal laser microscope are presented in Figure 2. Based on the results, different surface roughness was produced using different thermal spray processes. The T5-SHVOF coating showed the smoothest surface (Ra ~1.8 μm), and the roughest surface was observed in the TC-SPS coating (Ra ~9.2 μm).

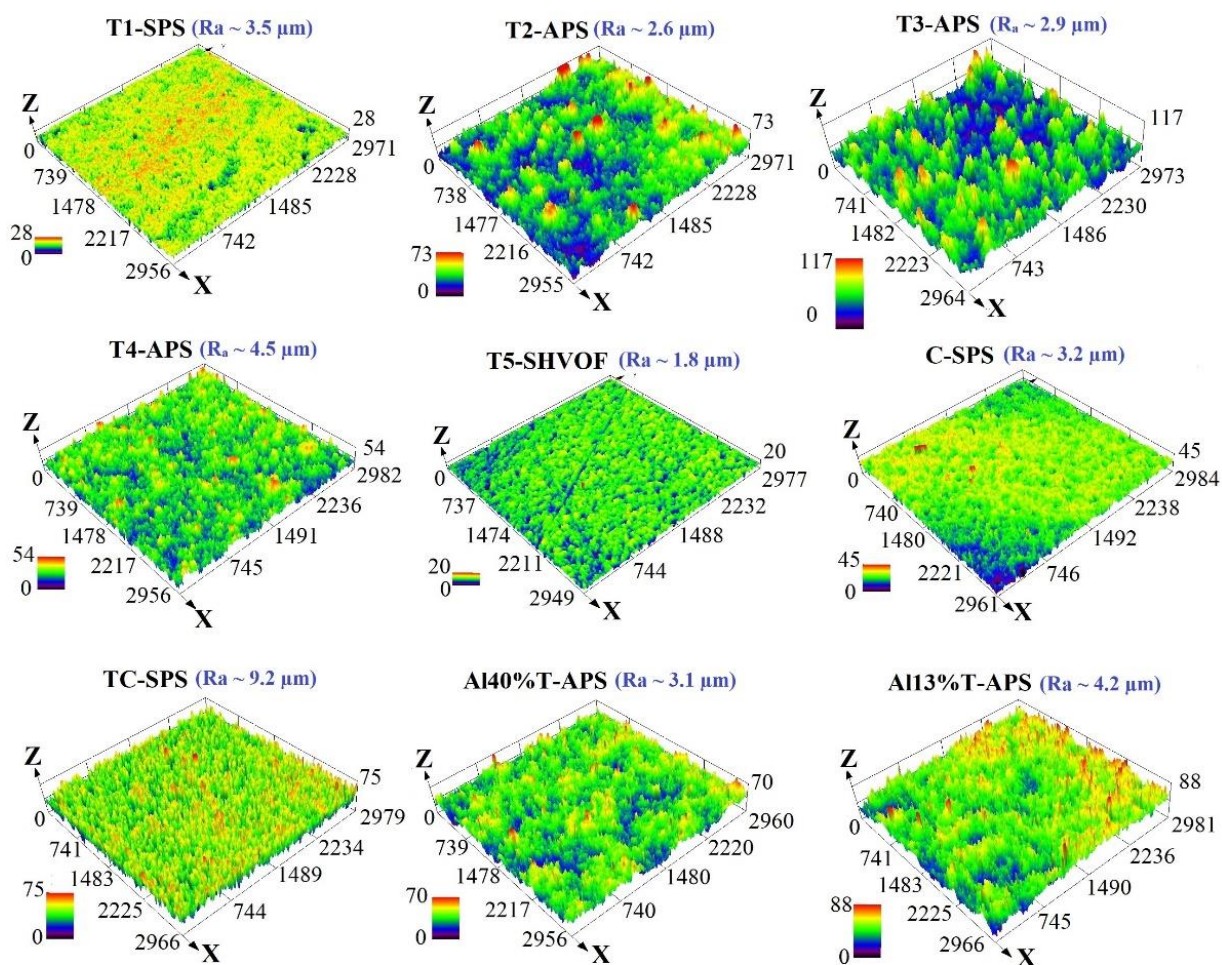

**Figure 2.** Top surface 3D images of the coatings and R$_a$ provided by the confocal laser microscope.

The SEM micrographs of the coating cross-sections and top surfaces are shown in Figures 3–7. Cross-sectioned SEM images of TiO$_2$ coatings in Figure 3 showed mechanically stable coatings well bonded to the surface. It is observed that the T1-SPS and T5-SHVOF have similar microstructural features. In T1-SPS and T5-SHVOF, the melted splats are

represented by the light gray zones, while the dark gray zones correspond to finely porous regions formed by unmelted particles from the feedstock suspension and resolidified particles [36,37]. In fact, the color difference between the light and dark gray zones corresponds to the presence of a large number of fine pores in the dark gray regions [38]. Additionally, the black zones are attributed to the larger pores. The origin of the unmelted particles in the T1-SPS coating could arise from the inflight particles traveling within the colder fringe of the plasma plume with lower thermal energy. Moreover, the relatively high vaporization enthalpy of water that was used as the solvent in the feedstock suspension would cool down the plasma jet and increase the ratio of unmelted particles in the coatings compared to APS coatings. The inflight particles in the HVOF process normally have lower temperatures than plasma processes, which, combined with using water as the solvent in the feedstock suspension, may result in preserving the unmelted particles in the T5-SHVOF coating. In addition, finer particles used in SPS and SHVOF processes are prone to resolidify faster within the spray jet and deposit on the surface [17,36,38]. The presence of agglomerated unmelted particles and resolidified particles may enhance the photocatalytic activity of the coatings by increasing the specific surface area of the photocatalyst and increasing the anatase content.

The surface view of the $TiO_2$ coatings is presented in Figure 4. It is observed that the asperities on the surface of T1-SPS and T5-SHVOF coatings are smaller compared to APS coatings, which could correspond to the finer feedstock powder that was used in the SPS and S-HVOF processes. Moreover, a high magnification view of the surface of T1-SPS and T5-SHVOF coatings in Figure 5 reveals the presence of a significant number of fine pores, unmelted agglomerated particles, and resolidified particles. In contrast, the surface of the APS coatings in Figure 4 shows the presence of mainly well-melted splats with limited porosity.

In Figure 3, the APS coatings also demonstrate comparable microstructures characterized by a typical lamellar arrangement. The lamellar structure results from the flattening of melted or partially melted inflight particles forming splats [39]. Other features that are observed in the APS coatings microstructure include the interlamellar pores and cracks and macro-pores [39,40].

SEM micrographs of C-SPS and TC-SPS are shown in Figure 6. A low plasma power was used to deposit the TC-SPS to maintain the $Cu_2O$ phase. The C-SPS demonstrates significantly smaller surface asperities compared to TC-SPS, resulting in a notable disparity in surface roughness between the two coatings, as illustrated in Figure 6. Moreover, the TC-SPS coating exhibits a highly porous microstructure.

Figure 7 shows the SEM micrographs of the cross-section and the surface of A40%T-APS and A13%T-APS coatings. Both coatings display surface features resembling the $TiO_2$ coatings deposited by the APS process, as depicted in Figure 4, with low porosity (black zones observed on the cross-sectional views). Additionally, in both coatings, the dark gray regions correspond to Al-rich areas, while the light gray regions are attributed to Ti-rich areas, as shown in the EDS images of the coatings in Figure 8. Moreover, The Cu-rich regions in TC-SPS are clearly depicted in the EDS images presented in Figure 8.

### 2.3. Antiviral Activity

2.3.1. $TiO_2$ Coatings

Figure 9 shows the antiviral activity of the $TiO_2$ coatings produced with SPS, APS, and S-HVOF processes under ambient light and UVA illumination. Overall, the antiviral activity of the coatings appears to be comparable or slightly enhanced under UVA light compared to ambient light, whereas T4-APS and T5-SHVOF show a rather higher activity level. Interestingly, the antiviral activity of the coatings under UVA and ambient light, regardless of the thermal spray production process, is comparable or slightly superior to that of the copper sample. Despite the minor fluctuation, which could correspond to the specific surface characteristics of the samples in this work, a consistent trend can be observed in the results.

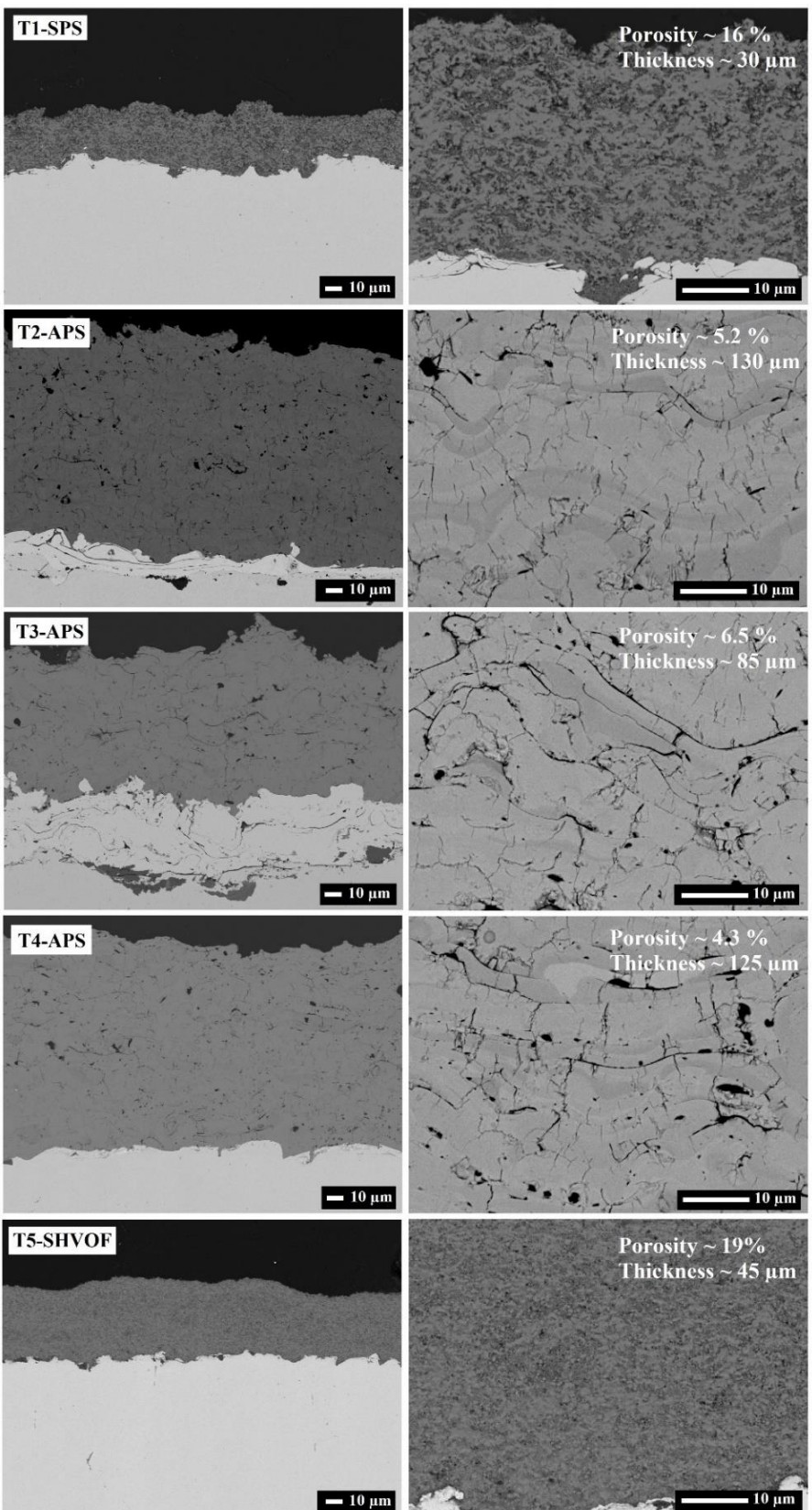

**Figure 3.** BSE FESEM micrographs of TiO$_2$ coatings' cross-sections at two magnifications.

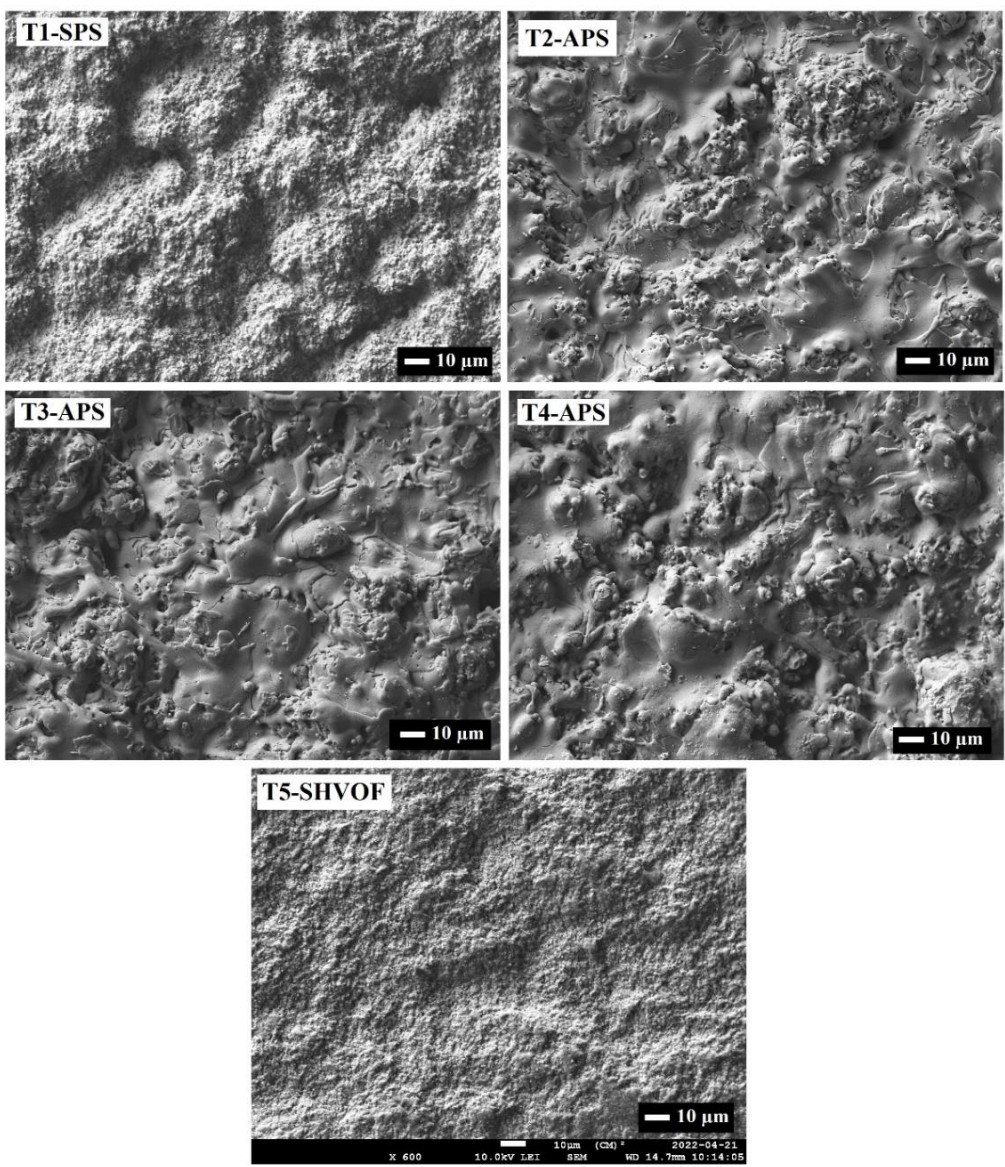

**Figure 4.** Top surface SE FESEM micrographs of TiO$_2$ coatings.

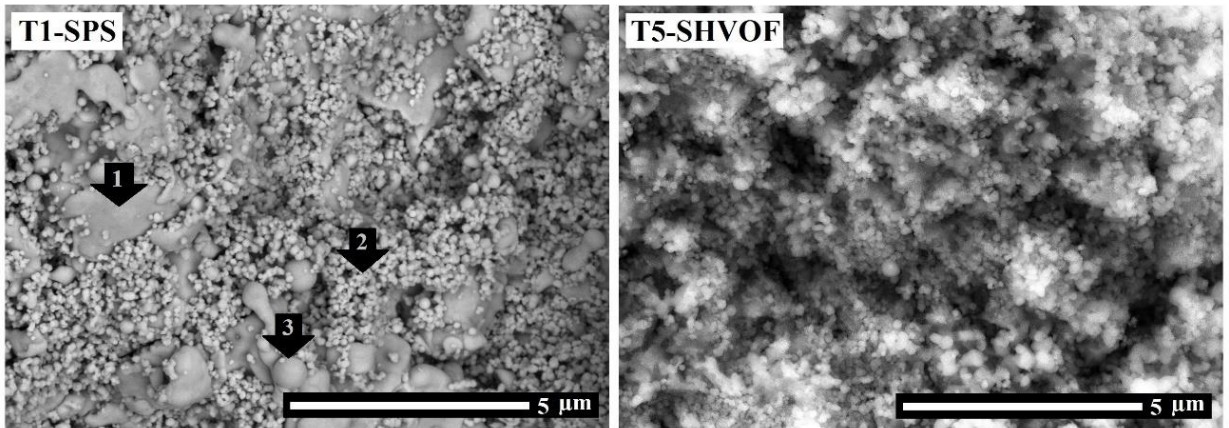

**Figure 5.** Top surface SE FESEM micrographs of T1-SPS and T5-SHVOF at high magnification, where (1) fully melted particles, (2) unmelted agglomerated particles, and (3) resolidified particles.

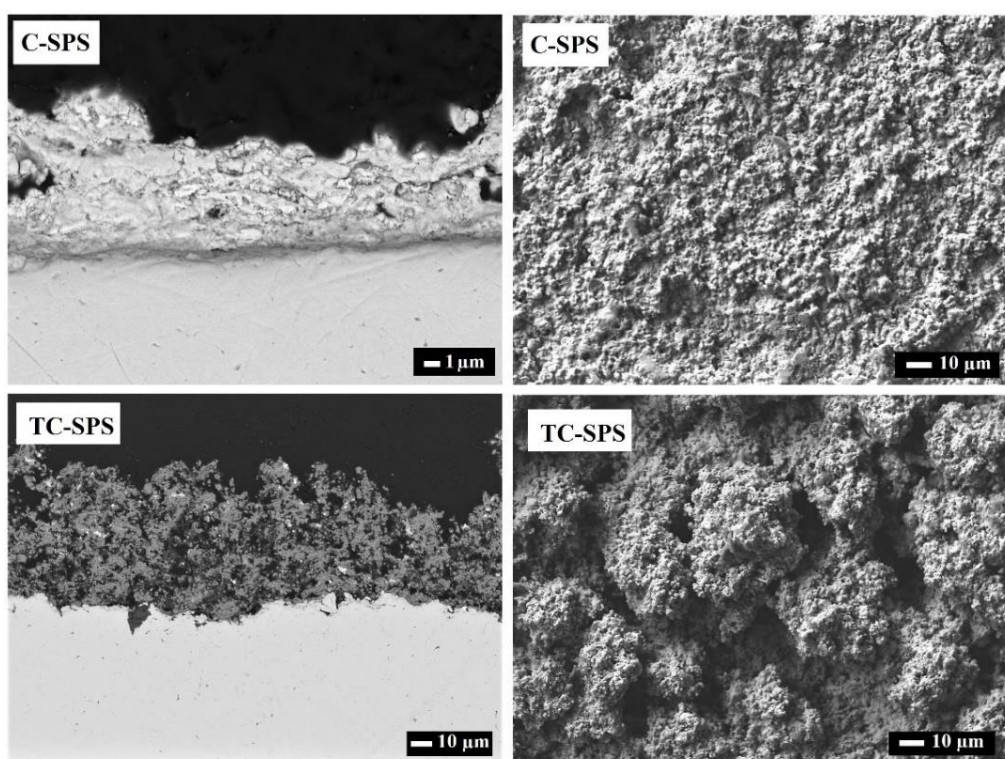

**Figure 6.** FESEM micrographs of C-SPS ($Cu_2O$) and TC-SPS ($TiO_2$-$Cu_2O$) coatings, left side: BSE cross-sectioned, right side: SE top surface images.

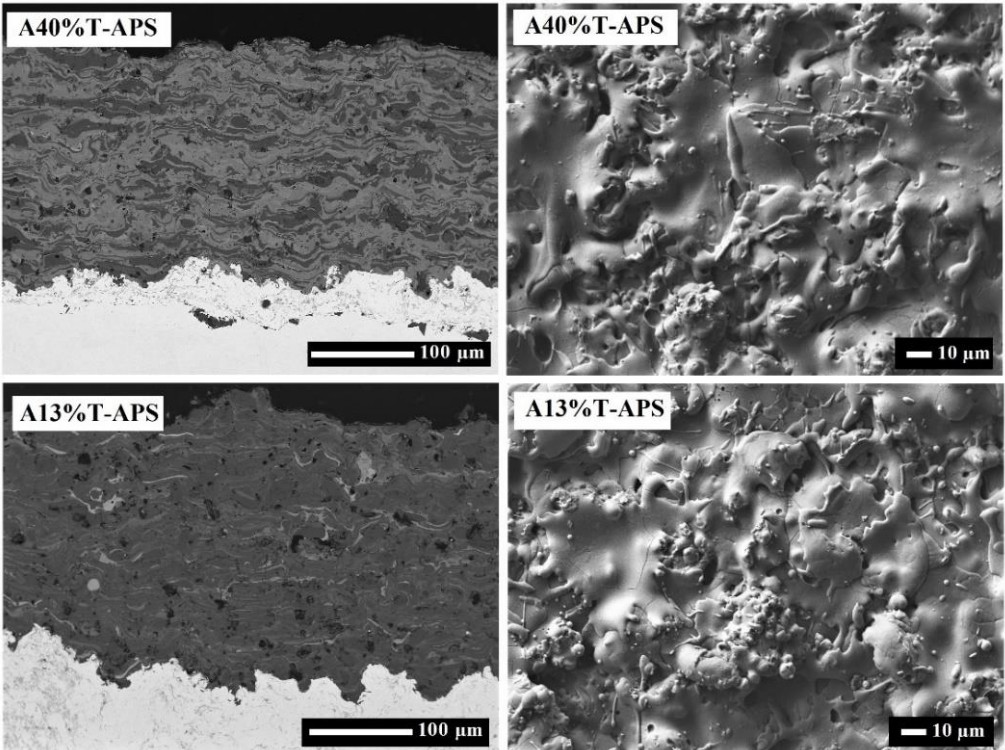

**Figure 7.** FESEM micrographs of $TiO_2$-$Al_2O_3$ coatings, left side: BSE cross-sectioned, right side: SE top surface images.

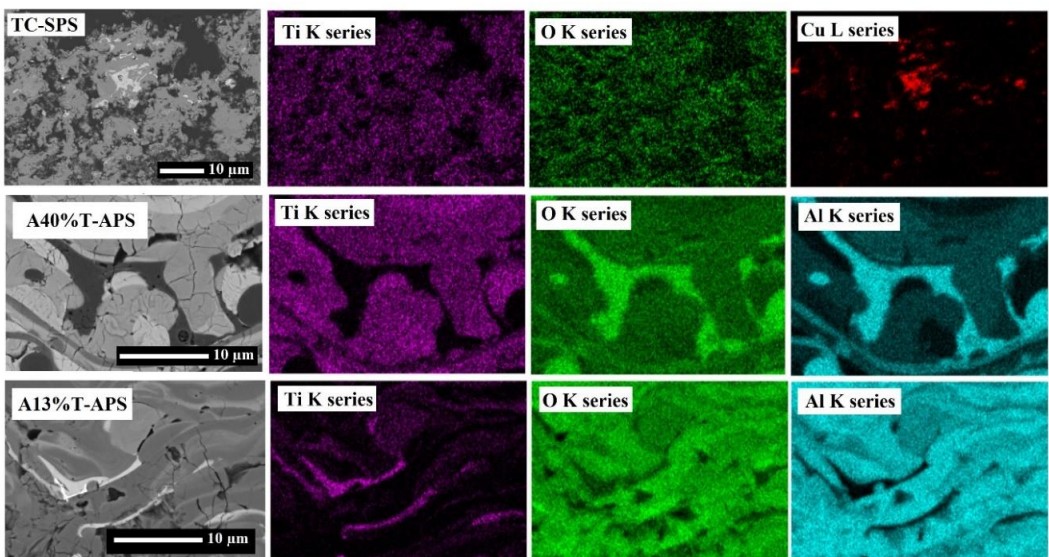

**Figure 8.** FESEM-EDX mapping of the cross-sectional view of TiO$_2$-Cu$_2$O and TiO$_2$-Al$_2$O$_3$ coatings.

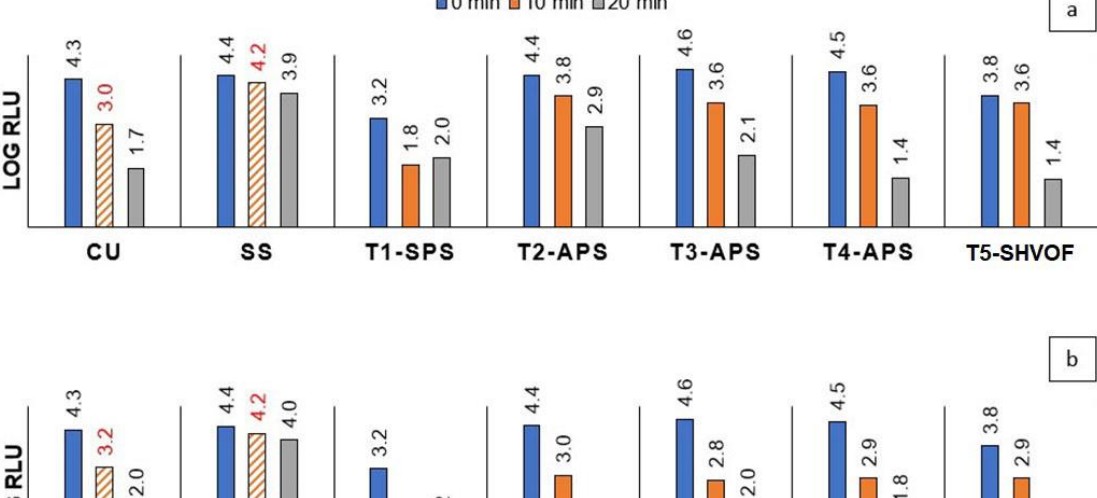

**Figure 9.** Antiviral activity assessment of TiO$_2$ coatings under (**a**) ambient light, (**b**) UVA light. Cu-10 min and SS-10 min were calculated by linear interpolation.

The antiviral activity of TiO$_2$ is due to the photocatalytic oxidation property [17,41]. When TiO$_2$ is exposed to photons with more energy than its bandgap, an electron (e$^-$) is excited from the valence band (VB) to the conduction band (CB), and a hole (h$^+$) is generated in the valence band. At the surface of the photocatalyst, these charge carriers interact with oxygen and water to produce highly reactive oxygen species (ROS) such as hydroxyl radicals (HO), O$_2^-$, HO$_2$, H$_2$O$_2$, etc. [42]. Equations (1)–(5) show the mechanisms of producing some ROS [42].

$$TiO_2 + h\nu \rightarrow TiO_2\ (e^- + h^+) \tag{1}$$

$$TiO_2\ (h^+) + H_2O \rightarrow TiO_2 + HO + H \tag{2}$$

$$TiO_2\ (e^-) + O_2 \rightarrow TiO_2 + O_2^- \tag{3}$$

$$O_2{}^- + H^+ \rightarrow HO_2 \tag{4}$$

$$O_2{}^- + HO_2 \rightarrow HO + O_2 + H_2O_2 \tag{5}$$

ROS such as $\cdot HO$, $O_2{}^-$, $HO_2\cdot$, and $H_2O_2$ have been suggested to be responsible for the degradation of viruses [41,42]. These ROS could damage the lipid membrane of the virus and degrade the capsid proteins or the envelope of the virus. Consequently, nucleic acid leakage occurs following virus destruction due to the decomposition of its genetic material [42,43].

Generally, the photogenerated electron–hole pairs in $TiO_2$ are produced under UV illumination due to their 3–3.2 eV bandgap energy [44]. However, thermal-sprayed coatings show photocatalytic activity under visible light due to the presence of oxygen vacancy and $Ti^{3+}$ ions in the coatings [33,37,45].

In general, anatase content in $TiO_2$ is believed to be the most important factor for photocatalytic activity [15,19,46]. However, other parameters, such as oxygen vacancy and surface morphology, could also play a critical role in the photocatalytic performance of thermally sprayed coatings [17,44].

It can be seen that after 20 min, T1-SPS and T5-SHVOF, with around 80% anatase, show rather similar or slightly lower Log RLU values than copper under both ambient and UVA illuminations. Notably, the Log RLU value of these two samples at 0 min, which determines the amount of the re-collectible virus from the coatings' surface, is lower than the other coatings and the dense control samples. The T1-SPS and T5-SHVOF samples have a relatively high surface porosity, as shown in Figure 5. This porosity allows a fraction of the virus to penetrate through the coatings almost immediately upon contact with the surface. The presence of sponge-like features in the coatings corresponding to the high surface porosity is not necessarily undesirable. This is due to the limited survival time of the coronavirus, which can last up to a few days [47]. Therefore, when the virus penetrates below the touchable section of the surface, it is unable to transmit infection and eventually decomposes. In addition, the high porosity and asperities on the surface create a larger photocatalytically reactive surface area, which, combined with the anatase content, could lead to a more efficient photocatalytic performance. Thus, in porous $TiO_2$ coatings, the antiviral activity depends on the combined effect of the efficient photocatalytic oxidation on the surface and displacement of the virus from the surface.

Likewise, in samples T2-APS, T3-APS, and T4-APS, the antiviral activity could be explained through a combined effect of several parameters. Figure 10 shows the color of the $TiO_2$ coatings produced with different thermal spray processes. It is seen that the APS coatings are much darker than SPS and S-HVOF coatings. It was shown previously that the darker the sub-stoichiometric $TiO_{2-x}$ coating in SPS coatings, the higher the oxygen vacancy content [17,33]. Furthermore, the level of darkness in sub-stoichiometric coatings is related to the concentration of the $Ti^{3+}$ ions [48–50].

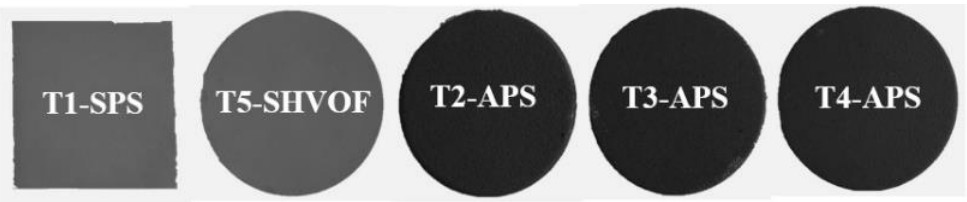

**Figure 10.** The color difference between $TiO_2$ coatings produced by various thermal spray processes.

In the thermal spray process, a high-temperature heat source is utilized to melt and accelerate feedstock material (such as powder or suspension) to form the coating. Oxygen vacancy could form during the intense heating of the particles, and its quantity is associated with the temperature of the in-flight particles. Notably, the presence of hydrogen as a reductive gas could promote the formation of oxygen vacancy [8,51,52]. Therefore, $Ti^{4+}$ could be reduced to $Ti^{3+}$ by accepting electrons from the reducing gas, and oxygen vacancies,

or even by receiving the photogenerated electrons under UVA and ambient lights [53]. Oxygen vacancies and $Ti^{3+}$ ions generate new energy levels below the conduction band, inhibiting charge carrier recombination and enhancing photocatalytic activity [54].

Consequently, the dark APS coatings may include higher levels of oxygen vacancy and $Ti^{3+}$ ions, resulting in the enhancement of antiviral performance. On the other hand, the lighter color in SPS and HVOF coatings could be due to a lower level of oxygen vacancy. Using water-based suspension feedstock for T1-SPS and T5-SHVOF would cool down the plasma jet due to the high vaporization enthalpy of water ($2.3 \times 106$ J kg$^{-1}$), decreasing the chance of producing oxygen vacancy in the coatings. Finally, it is worth mentioning that the synergetic effect between anatase and rutile in $TiO_2$ coatings may also promote photocatalytic reactivity by decreasing the recombination rate of the charge carriers [54].

The rather comparable antiviral activity for $TiO_2$ coatings under both UVA and ambient light could propose a more sustainable solution for indoor applications by using the commonly used visible light systems and for outdoor applications where the sunlight consists of around 53% visible light [55].

### 2.3.2. $TiO_2$-$Cu_2O$ and $Cu_2O$ Coatings

The suggested mechanism for virus deactivation by $Cu_2O$ is the direct contact of the virus with solid-state $Cu_2O$ compounds through valence-state Cu(I) species, resulting in the denaturation or degradation of its biomolecules. Therefore, $Cu_2O$ shows antiviral activity independent of optical absorption and in the dark environment [4,56]. The drawback lies in the oxidation of Cu(I) to Cu(II) under ambient conditions, which does not show significant antiviral activity [56]. In $TiO_2$-$Cu_2O$ composite, a combination of copper species with photogenerated holes in the valence band of $TiO_2$ under light irradiation can cause membrane damage, which is followed by protein oxidation and DNA degradation [28,29]. Furthermore, it was suggested that the photogenerated electrons in $TiO_2$ could be received by Cu(II), suppressing the self-oxidation of Cu(I) and ensuring a sustainable antiviral property of $Cu_2O$ [28].

Figure 11 shows the antiviral activity of the $Cu_2O$ and $TiO_2$-$Cu_2O$ coatings produced with SPS processes under ambient light, UVA illumination, and dark.

In Figure 11a,b, C-SPS and TC-SPS coatings demonstrate a relatively similar antiviral activity under both UVA and ambient lights, almost comparable to or slightly higher than that of the pure copper control sample. The Log RLU value of TC-SPS at 0 min shows that the quantity of re-collected virus was lower than C-SPS. The lower quantity of re-collected virus is due to the high porosity observed on the surface of TC-SPS shown in Figure 9. Therefore, the antiviral performance in TC-SPS could be linked to the combined effects of photocatalytic reactivity of $TiO_2$, penetration of the virus below the top surface of the coating due to porosity, and direct contact with $Cu_2O$. However, as seen in Figure 8, the Cu species are enveloped by $TiO_2$, which might have limited the antiviral activity due to the reduced direct contact with solid-state $Cu_2O$.

Figure 11c shows the antiviral activity of C-SPS and TC-SPS in dark conditions. It seems that the antiviral activity of C-SPS coating in the dark is lower than that that under UVA and ambient light. This outcome was not unexpected, since the direct-contact virus-killing mechanism in $Cu_2O$ is independent of photon energy [4,56].

Interestingly, at 20 min, the TC-SPS coating in dark conditions shows a significant decrease in the quantity of the viable virus, which is close to that of the copper sample. However, the Log RLU values for TC-SPS coating at 0, 10, and 20 min are somewhat similar. As mentioned earlier, this close similarity in Log RLU values indicates that the observed antiviral activity could correspond to penetration of the virus through the porous structure of the coating and its removal from the top surface. It can be assumed that optimizing the proportion of $Cu_2O$ in the $TiO_2$-$Cu_2O$ coating might probably be advantageous in terms of increasing the efficiency of antiviral activity in the dark.

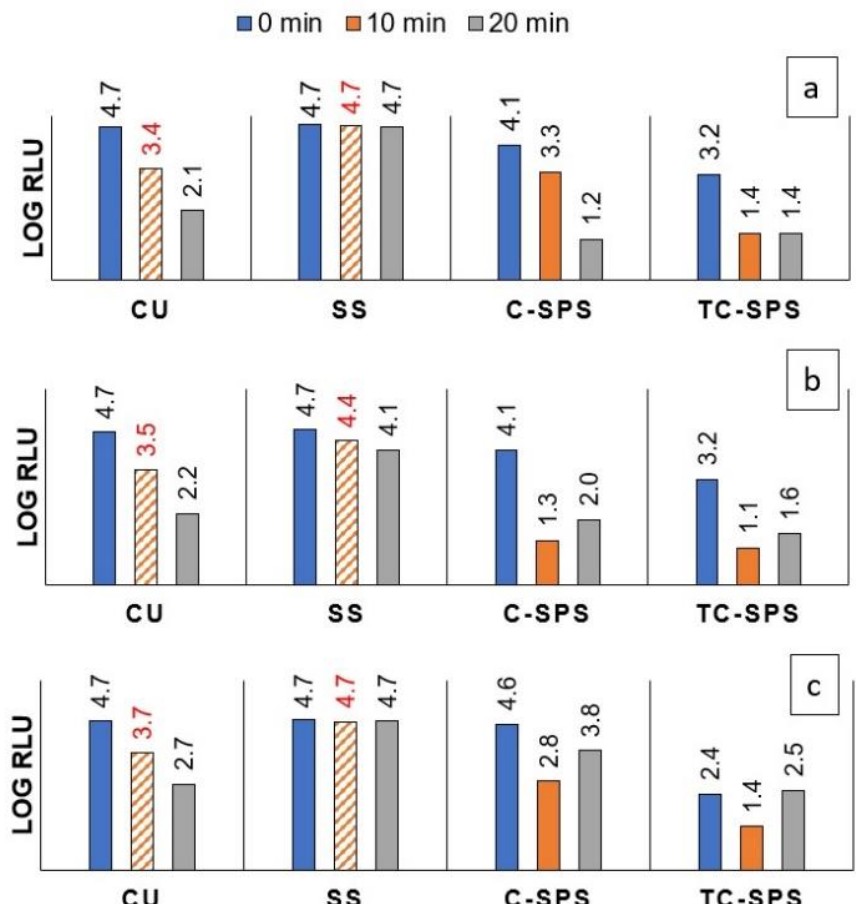

**Figure 11.** Antiviral activity assessment of $TiO_2$-$Cu_2O$ coatings under (**a**) ambient light, (**b**) UVA light, and (**c**) dark. Cu-10 min and SS-10 min were calculated by linear interpolation.

### 2.3.3. $Al_2O_3$-$TiO_2$ Coatings

Figure 12 shows the antiviral activity of A40%T-APS and A13%T-APS coatings under UVA and ambient light. The coatings showed almost no antiviral activity in both cases, which is similar to the stainless-steel control sample. In $Al_2O_3$-$TiO_2$ coatings, as shown in Section 3.1, most of the $TiO_2$ in the feedstock was transformed into $TiAl_2O_5$ in the coatings. Furthermore, the EDS images of $Al_2O_3$-$TiO_2$ coatings in Figure 8 showed the photocatalytically active surface of $TiO_2$ covered with inert Al-rich regions, resulting in the suppression of the photocatalytic activity in these coatings.

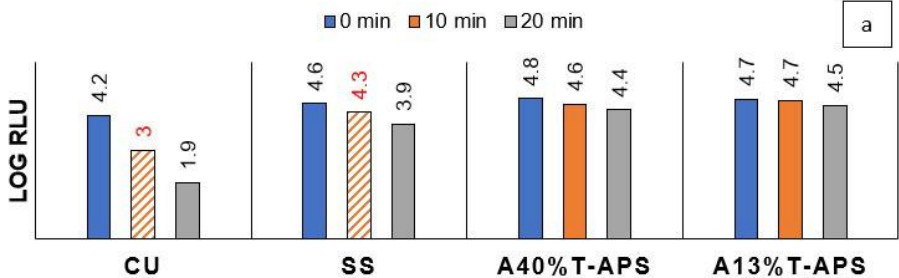

**Figure 12.** *Cont.*

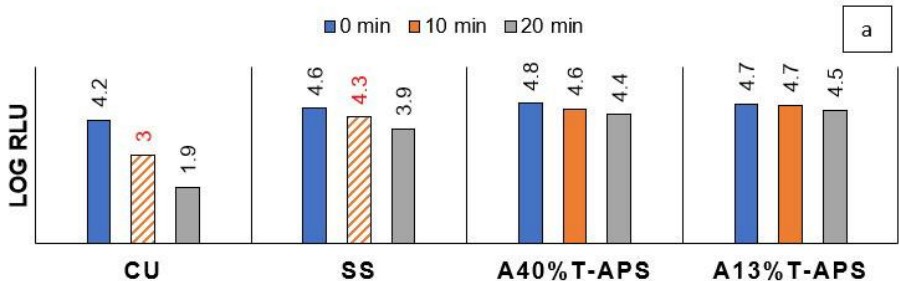

**Figure 12.** Antiviral activity assessment of $Al_2O_3$-$TiO_2$ coatings under (**a**) ambient light, (**b**) UVA light. Cu-10 min and SS-10 min were calculated by linear interpolation.

## 3. Materials and Methods

### 3.1. Preparation of Antiviral Coatings

Submicron-sized $TiO_2$ (TKB Trading, Oakland, CA, USA), micron-sized $TiO_2$ (Metco 102-Metco Oerlikon, Fort Saskatchewan, AB, Canada), nanostructured $TiO_2$ spray-dried (NEOXID T101 nano, Millidyne, Finland), $Cu_2O$ (PI-KEM, Tamworth, UK), $Al_2O_3$-40%$TiO_2$ (Amdry 6257-Metco Oerlikon, Fort Saskatchewan, AB, Canada), and $Al_2O_3$-13%$TiO_2$ (Metco 130-Metco Oerlikon, Fort Saskatchewan, AB, Canada) powders were used as feedstock materials to deposit coatings. SEM micrographs of the feedstock powders are shown in Figure 13. A diverse range of powder sizes is essential in the context of the different thermal spray processes. In the APS process, powders with particle sizes ranging from 10 to 100 µm are commonly used to avoid injection clogging [36]. On the other hand, sub-micron-sized powders are used in the SPS system.

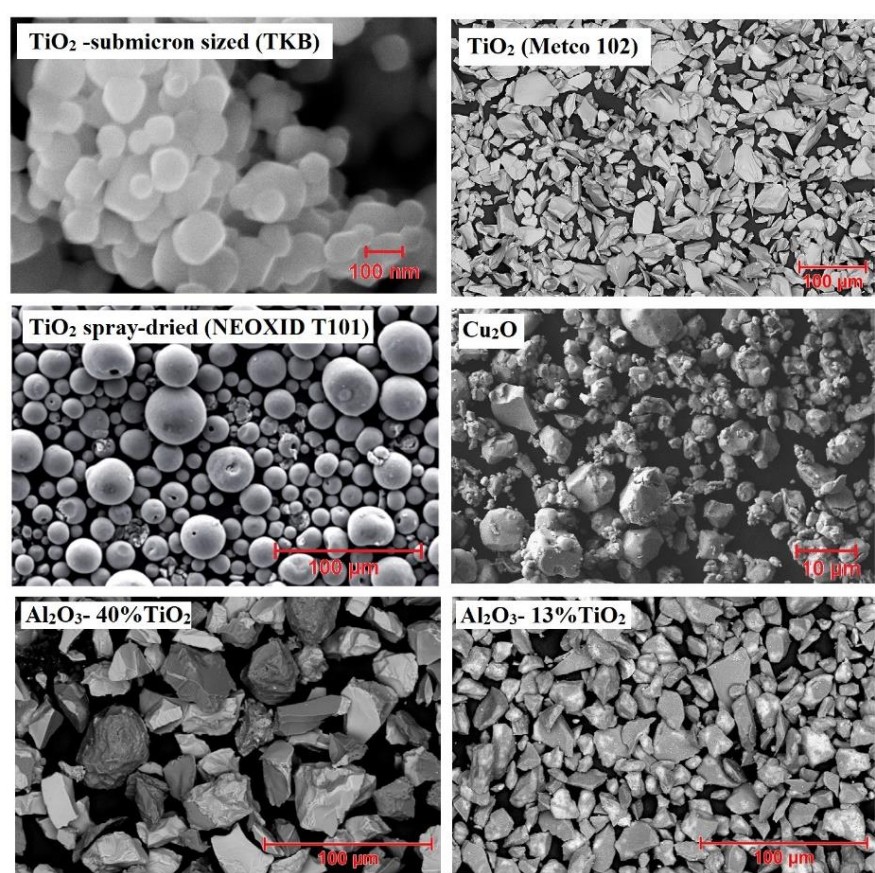

**Figure 13.** SEM micrographs of $TiO_2$ powder (Metco 102), $Al_2O_3$-40%$TiO_2$, and $Al_2O_3$-40%$TiO_2$ are taken from the data sheets provided by Metco Oerlikon.

Different thermal spray processes and systems were applied to produce the antiviral coatings on the grit-blasted 304 stainless steel coupons (1-inch diameter). The objective was to produce a range of coatings with different surface roughness, porosity, anatase content, and microstructures. To this end, three APS systems with low, medium, and high power were used. In addition, using the SPS process allows producing porous coatings with finer microstructures [57]. Furthermore, S-HVOF systems may preserve a higher fraction of anatase in the coatings due to the lower flame temperature [58]. The influence of various coating parameters is discussed in the following sections.

Details of the operating parameters are summarized in Tables 2 and 3. Five different types of $TiO_2$ coatings were produced to provide various microstructure, roughness, porosity, and anatase phase contents. The 10 wt.% water-based $TiO_2$ suspensions without any dispersing agent were used as the feedstock material to deposit T1-SPS and T5-SHVOF coatings by SPS and S-HVOF processes, respectively. An Axial IIITM APS torch (Northwest Mettech Corp., Surrey, BC, Canada) and ID-Nova Mk6 (Spraywerx technologies, North Vancouver, BC, Canada) HVOF system were used for producing the T1-SPS and T5-SHVOF coatings, respectively. ID-Nova is a compact hydrogen-fueled HVOF system designed to spray coatings onto the internal surfaces of the components. A hydrogen flow rate of 390 lpm and an oxygen flow rate of 190 lpm were used for the deposition of the T5-SHVOF coating. T2-APS, T3-APS, and T4-APS $TiO_2$ coatings were deposited by APS using SG-100 (Praxair Surface Technologies, Indianapolis, IN, USA), 100HE (Progressive Surface, Grand Rapids, MI, USA), and 3MB (Oerlikon Metco, Wohlen, Switzerland) plasma guns, respectively. The T2-APS, T4-APS, A40%T-APS, and A13%T-APS coatings were sprayed on the top of a bond coat to enhance the adhesion of the coating to the substrate. The presence of the bond coat does not affect the antiviral activity of the coatings.

**Table 2.** Thermal spray deposition parameters.

| Sample | Feedstock Material | Thermal Spray Process | Spray System | Powder Feed Rate (g/min) | Suspension Feed Rate (mL/min) | Current (A) | Spray Distance (mm) | Power (kW) |
|---|---|---|---|---|---|---|---|---|
| T1-SPS | $TiO_2$-TKB | SPS | Axial IIITM | - | 30 | 220 | 75 | 77 |
| T2-APS | $TiO_2$-Metco 102 | APS | SG-100 | 25 | - | 1000 | 100 | 35 |
| T3-APS | $TiO_2$-Metco 102 | APS | 100 HE | 100 | - | 400 | 115 | 95 |
| T4-APS | $TiO_2$-NEOXID T101 | APS | 3 MB | 20 | - | 500 | 75 | 29 |
| T5-SHVOF | $TiO_2$-TKB | S-HVOF | ID-Nova | - | 37 | - | 30 | - |
| C-SPS | $Cu_2O$-PI-KEM | SPS | 3MB | - | 35 | 500 | 50 | 17.5 |
| TC-SPS | $TiO_2$-25%$Cu_2O$-TKB | SPS | 3 MB | - | 35 | 500 | 50 | 20 |
| A40%T-APS | $Al_2O_3$- 40%$TiO_2$ (Amdry 6257) | APS | F4 | 30 | - | 480 | 120 | 30 |
| A13%T-APS | $Al_2O_3$- 13%$TiO_2$ (Metco 130) | APS | 100 HE | 90 | - | 260 | 115 | 65 |

$Cu_2O$ (C-SPS) and $TiO_2$-25%$Cu_2O$ (TC-SPS) coatings were deposited by the SPS method with a 3MB plasma torch. Here, 10 wt.% ethanol-based suspensions containing a small quantity of polyvinylpyrrolidone (PVP) (Sigma-Aldrich, Oakville, ON, Canada) (5 wt.% corresponding to the solid content) as a dispersing agent were used to deposit the coatings. A roller mill was utilized for five days to increase the homogeneity of the suspensions and to break down large agglomerates. The particle size distribution of the suspensions was measured by a Spraytec particle size analyzer unit (Malvern Instruments, Malvern, UK), as shown in Figure 14.

**Table 3.** Primary and secondary gas flow rates for plasma-sprayed APS and SPS coatings.

| Sample | Ar Flow Rate (L/min) | Secondary Gas Flow Rate (L/min) | | |
|---|---|---|---|---|
| | | H$_2$ | He | N$_2$ |
| T1-SPS | Gas flow (SLPM): 223, Gas mixture (%): Ar: 82/N$_2$: 11/H$_2$: 7 | | | |
| T2-APS | 42.7 | - | 27.4 | - |
| T3-APS | 85 | 66 | - | 66 |
| T4-APS | 40 | 8 | - | - |
| C-SPS | 50 | - | 10 | - |
| TC-SPS | 50 | 1 | - | - |
| A40%T-APS | 40 | 8 | - | - |
| A13%T-APS | 86 | 56 | - | 56 |

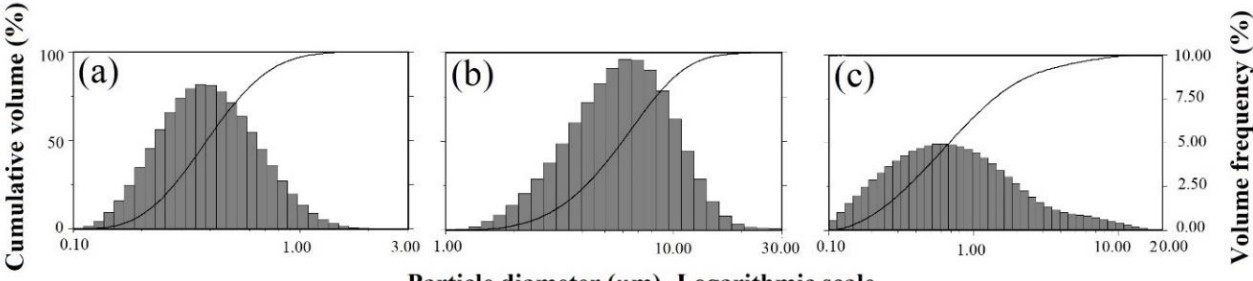

**Figure 14.** Particle size distribution of the suspensions: (**a**) TiO$_2$: d10 = 0.2 μm, d50 = 0.39 μm, d90 = 0.75 μm, (**b**) Cu$_2$O: d10 = 2.9 μm, d50 = 5.9 μm, d90 = 11 μm, (**c**) TiO$_2$-Cu$_2$O: d10 = 0.2 μm, d50 = 0.68 μm, d90 = 2.8 μm, to deposit T1-SPS/T5-SHVOF, C-SPS, and TC-SPS coatings, respectively.

To evaluate the role of the TiO$_2$ content on the antiviral activity of the coatings, TiO$_2$-Al$_2$O$_3$ composite coatings were manufactured by spraying two feedstock powders containing different quantities of TiO$_2$ (A40%T-APS: 40%TiO$_2$ and A13%T-APS: 13% TiO$_2$) using F4 (Oerlikon Metco, Wholen, Switzerland) and 100HE plasma torches.

### 3.2. Characterization of the Coatings

The morphology of the powders, the top surface of the coatings, and the polished cross-section of the coatings were observed with a field-emission scanning electron microscope (FESEM JSM 7600TFE, JEOL, Japan) in secondary electron and backscattered electron modes. The arithmetic average surface roughness (Ra) and topography of the coatings were evaluated with a confocal laser microscope (LEXT OLS4000 Olympus, Toronto, ON, Canada). The phase analysis of the coatings was carried out by the X-ray diffraction (XRD) technique (X'Pert pro, PANalytical, Philips, The Netherlands) in a range of 10−75° with Cu Kα radiation and a step size of 0.02°. The phase content was quantified by the tabulated reference intensity ratio using DIFFRACT.EVA software (Bruker, Billerica, MA, USA) and the PDF-2 database (international center for diffraction data).

### 3.3. Antiviral Activity Assessment

The antiviral performance of selected coatings was assessed in a containment level 2 bio lab under UVA light (Fisher Scientific, Canada), ambient light, and dark conditions. Two UVA (λ = 365 nm) lamps with a power of 15 W each with the illumination of around 500 lux were used. The intensity of UVA light was reduced by decreasing the transmission to 1% using 2.0 OD UV-NIR Neutral Density Filters (88-275, Edmund Optics Inc., Barrington, NJ, USA) to eliminate the impact of UVA illumination, killing the virus directly. The distance between the surface of the samples and UVA lights was around 7 cm, and the distance between the surface of the samples and the UV-NIR filters was around 2 cm, allowing normal air circulation over the surface of the samples. The ambient light was provided using white LED lights, which was generally used to illuminate inside the fume

hood. The intensity of the ambient light was measured at around 500 lux using a light meter (LANTEX LM-50KL, Woodbridge, ON, Canada).

The antiviral activity of the coatings was assessed using the HCoV-229E-Luc virus as a surrogate for the SARS-CoV-2 virus [59]. The HCoV-229E-Luc virus contains a Luciferase reporter gene allowing quantification of the relative amount of the viable virus in terms of relative luminescence unit (RLU), which was determined by measuring the luminescence signal using a Luciferase assay [60].

HCoV-229E-Luc stocks containing 1000 Plaque-Forming Units (PFUs) were prepared in Dulbecco's Modified Eagle Medium (DMEM) containing 2% Fetal Bovine Serum (FBS). First, 50 µL of the virus solution was put on the surface of the coatings. The coatings were exposed to ambient/UVA light or kept in the dark. The virus was recollected from the coating surface by pipetting 100 µL of media (Dulbecco's Modified Eagle Medium) at the coating's surface at three time points of 0 min, 10 min, and 20 min after exposure. A fresh coating was used for the test at each time point. The retrieved virus was then used to infect Huh7 cells in triplicates. Huh7 cells are human epithelial liver cell lines derived from hepatoma tissue, which supports HCoV-229E replication. A Renilla-Glo Luciferase Assay System (catalogue number E2720, Promega), with a Perkin Elmer plate reader (Ensight, Perkin Elmer), was used to quantify virus infection in terms of relative luminescence unit (RLU), and the logarithmic value of the average of the triplicate RLU measurements was reported. Furthermore, a copper plate (Cu), with high antiviral activity, and a stainless steel 304 plate (SS) with no antiviral activity were used as positive and negative control samples, respectively [61,62]. The antiviral activity of each coating can be determined by comparing its activity at 20 min to the activity of the same coating at 0 min. Subsequently, the antiviral activity of the coatings was compared to that of the control samples. For each day of the antiviral activity assessment experiment, independent control samples were used. The antiviral activity of the control samples followed a consistent trend during the whole duration of the experiment. To compare the results from the tests carried out on different days, the average values of the activity of the control samples were used. The schematic of the antiviral activity assessment process is shown in Figure 15.

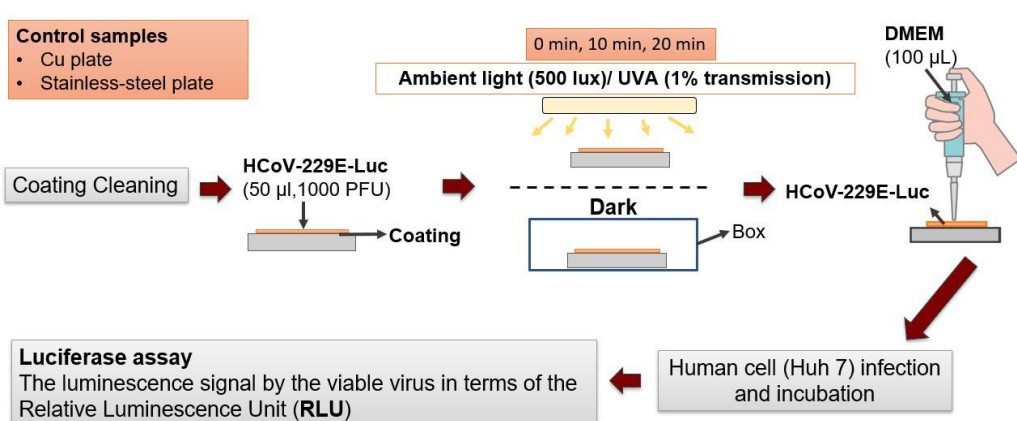

**Figure 15.** Schematic of the antiviral activity assessment process.

## 4. Conclusions

The antiviral activity of thermally sprayed coatings was assessed under ambient light, UVA, and dark conditions. $TiO_2$, $Cu_2O$, $TiO_2$-$Cu_2O$, and $Al_2O_3$-$TiO_2$ coatings were produced by APS, SPS, and S-HVOF processes. Regardless of the coating fabrication process, the $TiO_2$ coatings demonstrated relatively similar or somewhat enhanced antiviral activity compared to copper under visible light and UVA light. Furthermore, the $TiO_2$-$Cu_2O$ coating showed antiviral activity comparable to that of copper under visible light, UVA light. Results showed that the antiviral performance of the coatings corresponded to the collective influence of various parameters, including the photocatalytic activity of $TiO_2$, direct contact of $Cu_2O$ components, and surface properties of the coatings, such as

porosity and roughness. The surface porosity resulted in the transportation of a significant quantity of the virus beneath the touchable surface of the coatings, hindering its potential to cause infection. Although the regular cleaning processes of touched surfaces with high roughness and porosity could be challenging, an optimized level of surface properties could be beneficial to the total antiviral performance of the coatings through the combined effect of inactivating the virus and removing the virus from the surface.

Our results show that thermally sprayed coatings can offer a promising and potentially cost-effective approach for producing effective antiviral coatings using stable $TiO_2$. Other benefits include the absence of surface oxidation that occurs on copper, which can diminish its antiviral activity. The thermally sprayed coatings can be used on high-touch surfaces for indoor applications under ambient light, in the dark, and outdoor applications under sunlight.

**Author Contributions:** Conceptualization, C.M.; Data curation, E.A., H.K. and M.S.; Formal analysis, E.A., H.K., C.L. and C.M.; Funding acquisition, C.M.; Investigation, E.A., H.K., M.M.B. and M.S.; Methodology, E.A., H.K., M.S., C.L., R.S.L., J.O.B., M.A. and C.M.; Project administration, C.M.; Resources, C.L., S.M.S., R.S.L., J.O.B., M.A. and C.M.; Supervision, C.M.; Writing—original draft, E.A. and H.K.; Writing—review and editing, E.A., H.K., M.M.B., C.L., S.M.S., R.S.L., J.O.B., M.A. and C.M. All authors have read and agreed to the published version of the manuscript.

**Funding:** This study was financially supported by the Alliance Program of the Natural Sciences and Engineering Research Council of Canada (NSERC) and the Accelerate Program of Mitacs.

**Acknowledgments:** The authors extend their appreciation to David Kwan and Lan Huong Nguyen from Concordia University, Bruno M. H. Guerreiro from National Research Council Canada, Alan Burgess from Sprywerx Technologies, Steve Beaudin, Marc-Andre Ringuet, Adam Truchon, and Alexandre Gonçalves Andrade from Metal 7, Murray Pearson from Hatch, and Maurice Ringuette from University of Toronto.

**Conflicts of Interest:** The authors declare no conflict of interest.

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
