# Peer review of "A Comparative Study of the Antiviral Properties of Thermally Sprayed Coatings against Human Coronavirus HCoV-229E"

_catalysts, doi:10.3390/catal13071141_

Round 1

Reviewer 1 Report

This research paper entitled “A Comparative Study of the Antiviral Properties of Thermal Sprayed Coatings Against Human Coronavirus HCoV-229E” is an interesting and quite complete research paper on the antiviral coatings. The study conducted here has a real sense of applications not only for Coronavirus. TiO2 based coatings showed a good antiviral performance under UVA and ambient light.

Usually, this type of study uses TiO2 coating or Ag-doped TiO2 coating for their anti-bacterial and/or anti-Coronavirus properties. In this paper, the authors used TiO2 composites with other oxides, such as Cu2O and Al2O3. Even the authors demonstrated that TiO2-Cu2O coating showed antiviral activity comparable/similar to that of copper; the main advantage of those composites is the absence of surface oxidation that occurs on copper.

The paper is well organized and easy to read. As a fundamental research work with interesting results and detailed analysis, this paper can be accepted to be published in Catalysts after minor revision on typo errors.

Author Response

We would like to thank the reviewer for their careful evaluation of our article. We sincerely appreciate their time and effort in reviewing our work.

The manuscript has been updated to address any typographical errors, as suggested by the reviewer.

Reviewer 2 Report

Recommendations:

The efficacy of antiviral activity of all different coatings is in the range of 20% to 30% and some cases like in Alumina-TiO2 coating are negligible. Authors conclude that promising and effective. But readers do not see it as effective because efficacy is below 50%.  Authors should calculate the efficacy of the coating as a percentage to give a reader a clear message. 

Please revise the abstract with efficacy values to highlight the findings.  

Authors repeatedly mentioned that self-cleaning ability of the coating, however, there is no study included on the self-cleaning  ability for re-use the surface. Strongly recommended to include the results on regenerating the clean surface and effectiveness of cleaning the surface with light  to prove self-cleaning the surface.  This experiment will strength the manuscript further. 

Author Response

Comment 1) The efficacy of antiviral activity of all different coatings is in the range of 20% to 30%, and some cases like in Alumina-TiO2 coating are negligible. Authors conclude that promising and effective. But readers do not see it as effective because efficacy is below 50%. Authors should calculate the efficacy of the coating as a percentage to give a reader a clear message.

Our response:

We value the reviewer's comment, and we would like to provide further clarification. In our study, it was shown that the TiO2 coatings show rather similar or slightly better antiviral performance compared to copper. It is important to notice that the antiviral activity of each coating can be determined by comparing its activity to the activity of the same coating at 0 minutes. Subsequently, the antiviral activity of the thermal sprayed coatings was evaluated in comparison to copper, which is a recognized reference material for its antiviral properties. For instance, as shown in Figure 9, section 2.3.1, copper exhibited approximately 50% to 60% antiviral activity (reduction in viable virus). In comparison, the TiO2 coatings demonstrated activity ranging from around 40% to around 80% and, in one case, up to 100%. Showing rather similar up to around 50% more efficiency than copper.

With comparable antiviral performance to copper, the promising advantages of utilizing TiO2 as an antiviral material lies in the ability to apply a thin layer of coating using a less expensive material, as opposed to using bulk copper. Additionally, the thermal spray coating process offers a cost-effective manufacturing method, as mentioned in the introduction section (mainly in paragraph 7).

However, as mentioned in the manuscript, we would also like to emphasize to readers the impact of parameters such as surface morphology on the obtained results. We believe that our current approach of presenting the results allows for a better spotlight on this particular effect. Therefore, in this case, we would like to maintain the current method of presenting the antiviral activity, as it is also commonly used by other researchers when reporting the effectiveness of TiO2. We believe that this approach helps to enhance the readability and understanding of the manuscript for the readers. To provide additional clarity, we have updated section 3.3 accordingly and updated the Introduction.

Section 3.3 was updated to:  

Paragraph 3) The antiviral activity of each coating can be determined by comparing its activity at 20 min to the activity of the same coating at 0 minutes.

Section 1. Introduction, was updated to:

Paragraph 7)

In addition, historically, copper has been more expensive than TiO2. Furthermore, the price can vary depending on various factors, such as global demand, supply, economic conditions, and market dynamics. Thus, using copper-based materials on a large scale during a pandemic could be more costly.

.

Comment 2) Please revise the abstract with efficacy values to highlight the findings.

Our response:

Thanks for your comment. We updated the abstract to add more clarity.

The abstract was updated to:

Collectively, the thermally sprayed coatings showed comparable or slightly better antiviral activity compared to copper. The most significant level of activity observed was approximately 20% to 50% higher than that of a pure copper plate.

Comment 3) Authors repeatedly mentioned that self-cleaning ability of the coating, however, there is no study included on the self-cleaning ability for re-use the surface. Strongly recommended to include the results on regenerating the clean surface and the effectiveness of cleaning the surface with light to prove self-cleaning the surface. This experiment will strength the manuscript further.

Our response:

We would like to express our gratitude to the reviewer for their comment. We want to clarify that our article did not make any claims regarding the self-cleaning properties of the coatings. The term "self-cleaning" was mentioned once in the conclusion section. However, we understand that the reviewer might have been referring to the recyclability (reusability in terms of virus destruction) of the coating. We would like to highlight that evaluating the recyclability of the coatings' photocatalytic properties was not within the scope of our study. The assessment of the coatings' reusability will be investigated in the upcoming section of our research, where selected coatings will be applied to specific surfaces in public areas, such as airports. This phase will be carried out by one of our industrial partners. To prevent any confusion, we have removed the reference to the self-cleaning property of the TiO2 material that was mentioned in the conclusion.

Section4 was updated to:

Paragraph 2 ) “which inherently possesses self-cleaning characteristics” was removed, and the phrase was updated to: “Our results show that thermally sprayed coatings can offer a promising and potentially cost-effective approach for producing effective antiviral coatings using stable TiO2.”

Reviewer 3 Report

This is a very interesting work with great application potential.

The preparation and characterization of of the anti-virial surfaces has been described in detail, enabling them to be reproduced in another laboratory.  Thermal spray deposition parameters have been clearly defined and thoroughly discussed.

Evaluation of virucidal activity made with a virus HCoV-229E-Luc as a surrogate for the SARS-CoV-2 virus  using luminescent methods.  This qualitative method is quite sufficient to assess the virucidal activity of the tested coatings.

The manuscript is well organized and well written based on well-chosen literature.

Author Response

We would like to thank the reviewer for their careful evaluation of our article. We sincerely appreciate their time and effort in reviewing our work.

Reviewer 4 Report

In this work, the authors investigated the antiviral activity of thermally sprayed coatings of TiO2, Cu2O, TiO2-Cu2O, and Al2O3-TiO2 under ambient light, UVA, and dark conditions. Three coating processes were applied. Given the importance of surface contact in the transmission of COVID-19, this work could be considered for publication after revision. Below are some comments and suggestions for the authors’ consideration.

Line 66, one driving force for this work in the relatively high cost of copper. To reinforce this point, the price of copper and TiO2 could be provided.

The experimental setups need to be better justified. For example, three thermal spray processes were applied, but they were applied for the coating of different feedstock materials. By using such experimental parameters, the individual effect of thermal spray processes or feedstock materials cannot be revealed.

Line 189, the light intensity of the UVA should be provided in the unit of lux so that it can be compared with the ambient light (500 lux).

Line 371, the relationship of the coating color with the coating way and the antiviral ability observed in this study is different from those in literature. Provide some discussion if possible.

Line 418, TC-SPS showed better antiviral activity than pure copper under UVA, ambient lights, and dark conditions. This might be highlighted in the results and conclusions.

There are some typos in the manuscript. For example,

Line 169, it is “Figure 2” instead of “Figure 1”. The numbering of the figures hereinafter should also be revised.

Line 415, “[64].[65]” should be “[64][65].”

Line 428, “Figure 14 ©” should be “Figure 14 (c)”.

Author Response

Comment 1)  Line 66, one driving force for this work in the relatively high cost of copper. To reinforce this point, the price of copper and TiO2 could be provided.

Our responseWe thank the reviewer. Copper is historically more expensive than TiO2. An example of two of the potential feedstock powders used in thermal spray processes shows the price of TiO2 powder (https://tkbtrading.com/) to be about one-third of copper (https://www.supersonicspray.com/). However, these two materials are widely used and traded, and their price can vary depending on various factors such as global demand, supply, economic conditions, and market dynamics. Thus, the introduction section was updated to add more clarity.

Section 1. Introduction, was updated to:

Paragraph 7)

In addition, historically, copper has been more expensive than TiO2. Furthermore, the price can vary depending on various factors, such as global demand, supply, economic conditions, and market dynamics. Thus, using copper-based materials on a large scale during a pandemic could be more costly.

Comment 2) The experimental setups need to be better justified. For example, three thermal spray processes were applied, but they were applied for the coating of different feedstock materials. By using such experimental parameters, the individual effect of thermal spray processes or feedstock materials cannot be revealed.

Our response:

We express our appreciation to the reviewer for this comment. However, we would like to clarify that in thermal spray technology, the use of different feedstock powders is unavoidable due to the specific characteristics and limitations of each thermal spray process. Thus, the choice of feedstock material is dependent on the particular thermal spray process employed to generate the coatings.

The reason for using different feedstock materials was mentioned in section 3.1. Preparation of Antiviral Coatings, paragraph 1: “A diverse range of powder sizes is essential in the context of the different thermal spray processes. In the APS process, powders with particle sizes ranging from 10 to 100 µm are commonly used to avoid injection clogging [36]. On the other hand, sub-micron-sized powders are used in the SPS system.”

This section was highlighted in grey for the reviewer’s attention.

Comment 3) Line 189, the light intensity of the UVA should be provided in the unit of lux so that it can be compared with the ambient light (500lux).

Our response:

The illumination of the UVA light was measured at around 500 lux and was added to the description of the UVA light.

Section 3.3. Antiviral Activity Assessment was updated to:

Paragraph 1) Two UVA (λ = 365 nm) lamps with a power of 15 W each with the illumination of around 500 lux, were used.

Comment 4) Line 371, the relationship of the coating color with the coating way and the antiviral ability observed in this study is different from those in literature. Provide some discussion if possible.

Our response:

 We would like to thank the reviewer for this comment. The reference to the color of the coatings was addressed in our study. We discussed that while anatase content in TiO2 is commonly considered a crucial factor for photocatalytic activity, other parameters, such as oxygen vacancy and surface morphology, can also significantly influence the coatings' photocatalytic performance. In our research, we associated the dark color of the APS coatings with the presence of oxygen vacancies within the coatings. According to our previous findings cited in the manuscript, the darker the thermal spray coating is, the higher the oxygen vacancy content. Despite having a lower anatase content, these coatings exhibited photocatalytic activity under visible light due to the presence of oxygen vacancies. Consequently, even though the APS coatings contained less anatase compared to SPS and HVOF coatings, they demonstrated a comparable level of antiviral activity. The comprehensive explanation can be found in Section 2.3.1, where we thoroughly discuss these details and provide the corresponding references.

This section was highlighted in grey for the reviewer’s attention.

Comment 5) Line 418, TC-SPS showed better antiviral activity than pure copper under UVA, ambient lights, and dark conditions. This might be highlighted in the results and conclusions.

Our response

We value the reviewer's comment, and we would like to provide further clarification. In our study, it is important to notice that the antiviral activity of each coating can be determined by comparing its activity to the activity of the same coating at 0 minutes. Subsequently, the antiviral activity of the thermal sprayed coatings was evaluated in comparison to copper, which is a recognized reference material for its antiviral properties. Considering that both copper and TC-SPS show around the same level of activity under ambient and UV light. The activity of the TC-SPS in the dark was also discussed in section 2.3.2. As explained in the manuscript, for coatings such as TC-SPS, the influence of porosity needs to be also included in the analysis of the final results. We addressed the influence of porosity in detail in Sections 2.3.1 and 2.3.2 in order to provide a more comprehensive explanation. To provide additional clarity, we have updated the methodology section 3.2 accordingly.

 Section 3.2 was updated to:  

Paragraph 3) The antiviral activity of each coating can be determined by comparing its activity at 20 min to the activity of the same coating at 0 minutes.

Comment 6) Line 169, it is “Figure 2” instead of “Figure 1”. The numbering of the figures hereinafter should also be revised.

Our response: 

The 'Results and Discussion' was moved before 'Materials and Methods' following the journal's format, and the figure numbers were updated throughout the whole manuscript.

Comment 7) Line 415, “[64].[65]” should be “[64][65].”

Our response:

The typo was updated.

Section 2.3.2. TiO2-Cu2O and Cu2O Coatings was updated to:

Paragraph 1: “[64].[65]” was updated to “[64][65].”

Comment 8) Line 428, “Figure 14 ©” should be “Figure 14 (c)”.

Our response

The typo was updated.

Section 2.3.2. TiO2-Cu2O and Cu2O Coatings was updated to:

Paragraph 4: “Figure 14 ©” was updated to “Figure 14 (c)”.

Round 2

Reviewer 2 Report

The revised manuscript is suitable to publish in current form.